# Follow-Up Chest X-rays in Minor Chest Trauma with Fewer Than Three Rib Fractures: A Justifiable, Habitual Re-Imaging Industry?

**DOI:** 10.3390/healthcare10122471

**Published:** 2022-12-07

**Authors:** Amelie Deluca, Susanne Deininger, Florian Wichlas, Valeska Hofmann, Berndt Amelunxen, Julian Diepold, Thomas Freude, Christian Deininger

**Affiliations:** 1Institute of Tendon and Bone Regeneration, Spinal Cord Injury & Tissue Regeneration Center Salzburg, 5020 Salzburg, Austria; 2Department of Orthopedics and Traumatology, Salzburg University Hospital, Paracelsus Medical University, 5020 Salzburg, Austria; 3Department of Urology, Salzburg University Hospital, Paracelsus Medical University, 5020 Salzburg, Austria; 4Department of Trauma and Reconstructive Surgery, University of Tübingen, 72076 Tübingen, Germany

**Keywords:** rib fracture, X-ray, follow-up, control, chest trauma, thorax trauma

## Abstract

**Objective:** We evaluated the necessity of follow-up chest X-rays (CXRs) to exclude a pneumothorax after 1 week of initial hospital presentation in patients with no signs of respiratory distress and fewer than three rib fractures. **Materials and Methods:** Adult patients with fewer than three fractured ribs who presented at our Level I trauma center between 2015 and 2017 were evaluated retrospectively. Patients with sternal fractures, who had suffered a polytrauma, or were primarily treated with a chest tube were excluded. The patients’ and fractures’ characteristics, trauma mechanism, median follow-up time, and the number of required secondary interventions were recorded. **Results**: This study included 249 patients, 137 (55.0%) of whom were male, with a median age of 64.34 years. In 150 patients (60.2%) one rib was affected, in 99 patients (39.8%) two ribs were affected, with the fractured ribs being true ribs (1–7) in 72 cases (28.9%), false ribs (8–12) in 151 cases (60.6%), and both in 26 cases (10.4%). The affected thorax half was the left side in 124 cases (49.8%) and both thorax halves in 4 cases (1.6%). The median follow-up time was 9 ± 4 days. In the follow-up CXRs, six patients (1.6%) required delayed intervention (tube thoracostomy): one case of hemopneumothorax and five cases of pneumothorax. All of the patients fully recovered. **Conclusions**: Planned CXR follow-ups revealed only a small number of complications that needed intervention and therefore seem not to be necessary. Symptom-triggered reappearance seems to be more sufficient and economical compared to habitual reimaging.

## 1. Introduction

Rib fractures are commonly treated in the emergency department and observed throughout all ages with a higher incidence in geriatric patients due to osteoporosis and a weaker bone structure [1]. Hence, any rib fracture warrants a thorough evaluation of possible concomitant injuries, including spleen, liver, lungs, heart, kidney, and neurovascular structures [2].

Overall, fractures of fewer than three ribs (simple rib fractures) are usually less problematic than rib serial fractures but may still cause complications such as pneumothorax (PTX), delayed pneumothorax (DPTX), hemopneumothorax, or lung edema even after a few weeks’ time [3]. Some trauma centers regularly plan outpatient appointments after a simple rib fracture for patients to undergo a chest X-ray (CXR) follow-up appointment within one-week after initial trauma/presentation, even in the absence of respiratory manifestations. There are no general international guidelines in the literature, and it is being discussed, controversially, whether to carry out follow-up CXRs in every patient with fewer than three rib fractures [4,5]. Furthermore, there is no reference/evidence in the literature that the above practice leads to a better patient outcome.

The aim of this retrospective study is to determine whether a standardized control CXR in asymptomatic or low symptomatic patient is necessary and if it would reveal a significant number of required interventions or complications and therefore whether it should generally be recommended.

## 2. Materials and Methods

### 2.1. Study Design

This is a retrospective study of clinical and radiological outcomes in a cohort of 249 patients (≥18 years) with fewer than three acute rib fractures and no initial respiratory distress symptoms. All patients were managed initially non-operatively. Patients with sternal fractures and patients primarily treated with chest tubes were excluded.

Extracted data after management included: fracture characteristics, trauma mechanism, age, median follow-up time, number of required secondary interventions, associated injuries, and sex. Codes were used as the methods of study population selection to keep the participants anonymous.

### 2.2. Study Population

All patients included in this study presented at our Level I tertiary care trauma center between January 2015 and December 2017. Each patient was allocated to a group to simplify the overview according to the mechanism of injury (motor vehicle accident [MVA], work and sports accident, physical assault, fall at home), age (group I: 18–59 years; group II: 60–79 years; group III: ≥80 years), the number of fractured ribs (one or 2two and location of rib fractures (true ribs: 1st–7th, false ribs: 8th–12th ribs, both).

### 2.3. Radiological Fracture Evaluation

After the initial clinically based physical exam, a rib X-ray series of the affected site and CXRs were taken. The radiographs were evaluated by an independent radiologist and trauma surgeons using the hospital’s web-based picture archiving and communication system, which is integrated into the medical record system, IMPAX EE (Agfa Health Care, Bonn, Germany).

After confirming fewer than three rib fractures and excluding a PTX, a point-of-care ultrasonography as well as an E-FAST (extended focused assessment with sonography in trauma) scan were carried out to detect peritoneal- or pericardial fluid, subcutaneous emphysema, and/or hemothorax in a trauma patient.

The utility of a computed tomographic (CT) chest examination during initial evaluation has more importance in symptomatic and polytrauma patients (ISS > 16) and was not carried out in this study population. Patients with initial respiratory distress were not included.

### 2.4. Treatment Protocol

Patients who suffered fewer than three rib fractures, with no clinical signs of shortness of breath, and no PTX with no free peritoneal fluid were managed conservatively with pain medications. Every patient received an outpatient appointment between 5 and 10 days after initial presentation for a CXR follow-up. Follow-up monitoring was carried out in all patients. If the conventional CXR was not conclusive, or the patient complained about shortness of breath, an additional CT scan of the chest was carried out (*n* = 6).

### 2.5. Statistical Analysis

Collected data are expressed as mean ± standard deviation. Frequency distributions and summary statistics were calculated for demographic data points by using the χ^2^ test and Student’s *t* test. Statistical analysis was conducted using GraphPad Prism 9.0.0 (San Diego, CA, USA). The level of statistical significance was set at *p* < 0.05. This study included institutional-level linkage across two or more databases.

## 3. Results

This study included 249 consecutive patients aged 18 years or older (137 male [55%], mean age: 64.34 years; median: 68 years; range: 19 to 97 years).

The most common mechanism of injury was falls at home (70.7%). All data, including the characteristics of rib fractures, are summarized in Table 1.

The number of rib fractures, one (*p* = 0.0647) or two (*p* = 0.1257) broken ribs, did not differ between males and females. According to the age groups, the number of fractured ribs did not significantly vary among them: more than one fractured rib was observed in 33.7% of group I, in 43.1% in group II, and 43.9% in group III (*p* > 0.05). The association between all three age groups and the mechanism of injury is depicted in Figure 1. The most common mechanism of injury is a fall at home, especially as the age of the patient increases. Younger patients mainly sustain a simple rib fracture during sports accidents or an accident at work.

Associated injuries included clavicle bone fractures in three patients.

No patient was lost during follow-up and hence excluded. The median follow-up time was 9 ± 4 days.

There were no missed PTXs on the initial CXR as evaluated by a radiologist and trauma surgeon.

A delayed intervention was required in six patients (1.6%) upon CXR follow-up approximately 1 week post trauma: one case of hemopneumothorax (age 92) and five cases of PTX > 2 cm (3 in age group II, 2 in group III; median age 68 years). The PTX was not present on the initial CXR. Some patients (65%) presented with respiratory distress. Only one patient returned to the ED prior to the official follow-up appointment due to respiratory distress. All patients ended up receiving a tube thoracostomy for drainage due to the progression of respiratory distress and a pneumothorax > 2 cm. The mean duration of chest tube intubation was 4.5 days and all patients fully recovered. The patients were followed-up with CXRs. The mechanism of injury included four falls at home, one physical assault and one sports accident. The univariate analysis showed that there was no significant difference between the number of fractured ribs or the patients’ age and the development of a PTX.

The above protocol has been summarized in a flow chart and represents an algorithm derived from our findings (Figure 2).

## 4. Discussion

The incidence of a delayed pneumothorax (DPTX) due to minor chest trauma occurs in less than 2% of patients according to the literature [6]. Those results are concurrent with the results of this present study. Only 1.6% of our patients presented with a DPTX and only one patient had severe respiratory distress.

A limitation of this study, and diagnosing rib fractures in general, was/is the usage of only native X-ray analysis. However, even the study by Liu et al. showed that the initial detection of rib fractures in the costochondral junction was only diagnosed in 60% of patients by using a CT scan. Most missed rib fractures were accompanied by visible fractures on the same or adjacent first to second rib [7].

When isolated, rib fractures have a relatively low morbidity and mortality, and failure to detect isolated rib fractures does not necessarily alter overall patient management or outcomes in uncomplicated cases [8] in patients with minor trauma. It is possible that some occult PTXs were overlooked in our follow-up CXRs, as they would only be visible on CT imaging, but apparently the patients did not present with any symptoms.

Therefore, thoracic CT scans are the “gold standard” for early detection of a PTX and they are the imaging modality of choice for seriously injured blunt trauma patients and patients with respiratory distress [9]. For uncomplicated cases and patients with no shortness of breath, utilizing E-FAST technologies is of high value to exclude PTX and abdominal injuries [10].

The association between age groups and mechanism of injury shows that mainly elderly people (age group III) sustain rib fractures due to a fall at home. This may of course raise an alert around public health at home and domestic safety. Western society is showing a general increase in geriatric trauma patients. The number of falls occurring in an in-home setting and resulting in a rib fracture is indeed high. Fall prevention programs as well as 24-h caregivers and other social services have existed for several years and continue to expand. The goal of the local population is generally to be able to remain in their own home environment if possible. Moving to a retirement home serves as an ultima ratio. In addition, there is often a lack of space for available care in such homes. In general, this is a problem that will increasingly affect our aging society in the coming years [11,12]

The limitations of this study include the small number of patients, absence of a control group, and variables such as CXR interpretation by different physicians. However, we believe that the obtained data and knowledge are still valuable.

We can conclude that planned CXR follow-ups after minor chest trauma with fewer than three fractured ribs can be omitted. Symptom-triggered patient reappearance, such as respiratory distress or abdominal pain after rib fractures, in hospitals is the standard of care in many centers and seems to be sufficient and more economical compared to regular reimaging. However, it is important to inform and educate every patient about possible side effects and the need for immediate follow-up if symptoms of respiratory distress appear.

## Figures and Tables

**Figure 1 healthcare-10-02471-f001:**
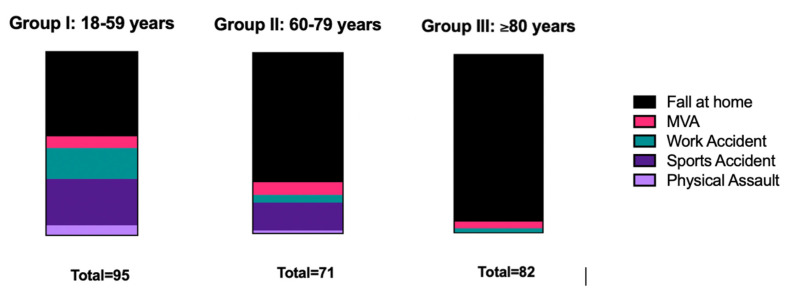
Association between age groups (I to III) and mechanism of injury.

**Figure 2 healthcare-10-02471-f002:**
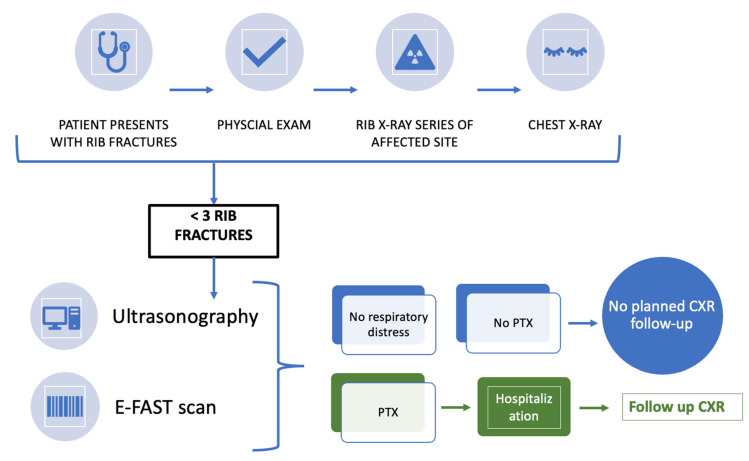
Flow chart of minor chest trauma presenting fewer than three rib fractures. PTX = pneumothorax; CXR—chest X-ray.

**Table 1 healthcare-10-02471-t001:** Clinical characteristics of trauma patients with <3 rib fractures.

MECHANISM OF INJURY	*N* (%)
MOTOR VEHICLE ACCIDENT	30 (12.0)
ACCIDENTS AT WORK	18 (7.2)
SPORTS ACCIDENTS	19 (7.6)
PHYSICAL ALTERCATION	6 (2.4)
FALL AT HOME	176 (70.7)
**AGE**	
**GROUP I: 18–59 YEARS**	95 (38.2)
**GROUP II: 60–79 YEARS**	72 (28.9)
**GROUP III:** **≥** **80 YEARS)**	82 (32.9)
**NUMBER OF FRACTURED RIBS**	
1	150 (60.2)
2	99 (39.8)
**LATERALITY**	
LEFT SIDE	124 (49.8)
RIGHT SIDE	121 (48.6)
BILATERAL	4 (1.6)
**LOCATION**	
TRUE RIBS (I–VII)	72 (28.9)
FALSE RIBS (VIII–XII)	151 (60.6)
BOTH	26 (10.4)

## Data Availability

All data can be obtained from the authors.

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
