# Peer review of "Follow-Up Chest X-rays in Minor Chest Trauma with Fewer Than Three Rib Fractures: A Justifiable, Habitual Re-Imaging Industry?"

_healthcare, 2022, doi:10.3390/healthcare10122471_

Round 1

Reviewer 1 Report

Manuscript title: Follow-up Chest X-rays in minor chest trauma with less than 3 rib fractures: a justifiable, habitual re-imaging industry?

Comments

This manuscript reports a study designed to determine whether a standardized control chest X-ray (CXR) in a- or low symptomatic patients is necessary and if it would reveal a significant number of required interventions or complications. The Introduction section includes a clear rationale for the study. The study aims can be addressed using the selected method (retrospective study). Overall, the reporting of the study could benefit from following some guidelines. Also, statistical analysis and results required some additional information and possibly major revision. Please find specific comments below. Pages are numbered according to the PDF file generated for review.

Major comments

1. Consider using reporting guidelines from the EQUATOR Network to ensure all relevant information is presented. Being a retrospective study of medical records, the authors are directed to the RECORD guideline (https://www.equator-network.org/reporting-guidelines/record/).

2. Lines 63-67. It is unclear why patients were allocated into groups based on criteria other than the number of fractures (i.e., the main factor being analyzed here). Nonetheless, Table 1 shows all patients without any grouping where I would expect results of the groups <3 and  ³3 besides the overall summaries. Figure 1 display age subgroups (18-59 y, 60-79 y, ³80 y) that were not justified in the Methods section and again ignored the remaining posed criteria.

3. Lines 101-102. In the sentence: ‘There was no significant difference between male and female with 1 (P=0.0647) or 2 101 rib fractures (P=0.1257)’: What variable was compared between males and females? To what comparison does the P-value = 0.1257 refer?

4. Line 111. Again, data is compared between age groups. How does it help answer the research questions?

5. Figure 2. The purpose of the flowchart is unclear. Does it represent the flow of patients at each study phase (in which case the number of participants at each stage is missing) or an algorithm derived from your findings?

Minor comments

1. Lines 91-92. Funding and ethical approval info seem misplaced. Please double-check the instructions for the authors of the journal.

2. Line 106. Sector plots are not the best choice to represent an association. Consider bar plots (or tables) instead. Also, ‘correlation’ represents the relationship between scale variables and should not be used here.

Author Response

Thank you very much for your comments and suggestions. We adapted them.

Kind regards

The authors

Reviewer 2 Report

Thank you for the opportunity to review the paper entitled: “Follow-up Chest X-rays in minor chest trauma with less than 3 rib fractures: a justifiable, habitual re-imaging industry?” by Deluca and colleagues. I enjoyed reading this paper. 

The authors describe a retrospective description of 249 patients reevaluated for pneumothorax one week following blunt thoracic trauma. I congratulate the authors for a well-written, concise and appropriately abbreviated paper. 

I do have some comments though. 

1.    Surgeons generally attend to thoracic trauma in trauma centers, therefore why are the authors from a tendon and bone regeneration and tissue center, orthopedics, reconstructive and urology departments? Please explain since although this paper is interesting, the absence of surgery in this paper demerits the credibility and strength of the paper because of the lack of intellectual authority provided by a surgeon. 

2.    Another question is regarding the number of patients having presented blunt thoracic trauma as a result of falling at home during a single period of only two years at a single center. This raises an alert as to the public health at home and domestic safety.

3.    I appreciate the comment of line 118 detailing that no difference was found regarding number of rib fractures and the presence of pneumothorax as this is a valuable data. 

4.    I applaud the authors for the flow chart, I would like to see however the management branch of those with more than 3 rib fractures. 

5.    Although the authors politely address the limitation of plane radiography on line 129, I disagree. Simple chest x rays are more than enough to rule out rib fractures. Therefore, this is not a limitation to the study but I thank the authors for the remark.

Overall this is a great paper, short, and easy to read, the authors should take the recommendations into consideration. The value of the information provided is useful for future clinical follow-up, however I must insist that the absence of a surgical division as primary authors highly limits the intellectual authority as none of the authors come from a general surgery or thoracic surgery department. As a result, this will affect readership severely and the credibility of this paper.  

Author Response

(The authors gave the same response as above.)

Round 2

Reviewer 1 Report

Thank you for the opportunity to discuss your report. All comments were properly addressed. I have no new comments.

Author Response

Thank you very much for you comments.

Kind regards

The authors

Reviewer 2 Report

the authors have appropriately answered all my questions

Author Response

Dear Reviewer,

Thank you for your positive review of our work.

With collegial greetings

The authors
